# Thulium Laser Vapoenucleation of the Prostate (ThuVEP) in Men at High Cardiovascular Risk and on Antithrombotic Therapy: A Single-Center Experience

**DOI:** 10.3390/jcm9040917

**Published:** 2020-03-27

**Authors:** Daniele Castellani, Mirko Di Rosa, Luca Gasparri, Michele Pucci, Marco Dellabella

**Affiliations:** 1Department of Urology, IRCCS INRCA, 60127 Ancona, Italy; l.gasparri@inrca.it (L.G.); m.pucci2@inrca.it (M.P.); m.dellabella@inrca.it (M.D.); 2Unit of Geriatric Pharmacoepidemiology, IRCCS INRCA, 60127 Ancona, Italy; m.dirosa@inrca.it

**Keywords:** aging, anticoagulants, benign prostatic hyperplasia, intraoperative complications, laser therapy, platelet aggregation inhibitors, postoperative complications, prostatectomy, ThuVEP

## Abstract

Bleeding is the most common complication of transurethral resection of the prostate and simple open prostatectomy, especially in men on antiplatelet/anticoagulant therapy. The present study aimed to evaluate the safety and effectiveness of thulium laser vapoenucleation of the prostate (ThuVEP) for benign prostatic hyperplasia in patients on chronic antithrombotic medications. Between January 2015 and December 2019, 88 men underwent the procedure under antithrombotic agents in our center. The mean age was 74.7 ± 6.1 years. Median prostate volume was 66.5 mL. Patients on oral anticoagulants were bridged to low-molecular-weight heparin (*n* = 35). Aspirin (*n* = 39), clopidogrel (*n* = 10), and ticlopidine (*n* = 4) were maintained. Of the patients, 69.3% had an American Society of Anesthesiologists score ≥ 3. Blood loss at 24 h was comparable in all groups. Median catheterization length and postoperative stays were 2 and 3 days respectively. Acute cardiovascular events occurred in 2 patients (2.3%). Of the patients, 4 required prolonged bladder irrigation, 2 required blood transfusions, 1 required a cystoscopy for bleeding control, and 1 required a suprapubic cystostomy for blood clot evacuation. No patients died within 30 days of being discharged. Late complications occurred in 3 (3.8%) patients (1 optical urethrotomy and 1 bladder neck incision for stenosis; 1 acute myocardial infarction). All follow-up visits (1, 6, and 12-month) showed a significant improvement in all urinary parameters compared to baseline. ThuVEP appears to be a feasible surgical option in high-risk patients on antithrombotic regimens, with acceptable postoperative morbidity, good functional outcome, and low incidence of medium-term reoperation rate.

## 1. Introduction

Traditionally, transurethral resection of the prostate (TURP) and simple open prostatectomy (OP) have been considered the gold standard for the surgical treatment of clinical benign prostatic hyperplasia (BPH), depending on gland volume. Intraoperative and postoperative bleeding is the most common complication of TURP and OP, usually leading to blood transfusions, prolonged hospitalization, and reoperation, especially in men on anticoagulant (AC)/antiplatelet (AP) treatments [1,2,3]. For many decades, urologists would interrupt AC/AP medications in the provision of surgery. However, antithrombotic therapy should not be discontinued in patients at risk because withdrawal can lead to thromboembolic events in the postoperative period [4,5]. Lasers have been introduced in clinical practice in order to overcome the morbidity of TURP and OP and have been demonstrated to be superior in blood loss and transfusion rates in comparison to traditional surgery [6,7,8,9]. According to the current European Association of Urology (EAU) guidelines, photoselective vaporization of the prostate (PVP) using a laser is considered the standard/first-choice surgical therapy in men who cannot stop AC/AP treatment [10]. Both the EAU and American Urological Association guidelines recommend that transurethral anatomical enucleation of the prostate with holmium (HoLEP) or thulium laser support can also be offered to patients who are at higher risk of bleeding [10,11]. Thulium laser vapoenucleation of the prostate (ThuVEP) has been demonstrated to be a safe and size-independent procedure, with low perioperative morbidity, even in large-volume prostates, and also to produce long-term improvement in patient-reported outcomes and objective voiding parameters [12]. Nevertheless, there are few papers in the literature focusing on its safety and effectiveness in men on AC/AP [13,14,15]. Currently, there is no data on the impact of different AC/AP regimens on complications of ThuVEP in this population.

This study aimed to evaluate the complication rate and functional results of ThuVEP in patients who cannot interrupt antithrombotic therapy. The secondary purpose was to investigate the relationship between different antithrombotic regimens and the outcomes.

## 2. Experimental Section

### 2.1. Study Design and Participants

We retrospectively analyzed data from all men who consecutively underwent ThuVEP at our center. The inclusion criteria were: lower urinary tract symptoms (LUTS) non-responsive to medical therapy, an International Prostate Symptom Score (IPSS) ≥ 8, maximal urine flow rate (Qmax) less than 15 mL/s, absolute indications for surgery according to EAU guidelines, and cardiovascular diseases that required continued use of AC/AP medications. A multidisciplinary approach involving urologists, cardiologists, and neurologists was employed in managing all patients. No procedures were performed within one year of the coronary stent placement according to current practice [5]. Novel oral anticoagulant agents and warfarin were stopped 2–3 days before surgery and bridged with low-molecular-weight heparin (LMWH) [5]. The dose of LMWH was decreased in patients with impaired renal function. Oral AC was resumed until the bleeding was almost completely subsided [5]. Exclusion criteria were: urethral stricture, prostate cancer, previous pelvic irradiation, neurogenic bladder, previous urethral/prostatic surgery, and concomitant lower urinary tract surgery (urethrotomy, lithotripsy, and resection of bladder tumor). Suspicious prostate cancer was ruled out with prostate biopsy. All patients who underwent ThuVEP and met the inclusion criteria were asked to take part in the study. Preoperative prostate-specific antigen (PSA), transrectal ultrasound measurement of prostate volume (PV), presence of an indwelling catheter, Qmax, and self-administrated IPSS, including the quality of life (QoL) index and American Society of Anesthesiologists (ASA) score on physical status classification system, were gathered. The decrease in hemoglobin (Hb) level was measured by evaluating Hb before and 24 h after surgery. Postoperative early complications were considered within 30 postoperative days and were reported according to the modified Clavien classification system [16]. Follow-up visits were scheduled at 1, 6, and 12 months after surgery. The investigation was carried out following the rules of the Declaration of Helsinki. The study has been approved by our local Ethical Board (number DGEN 421/2017). Written informed consent was obtained from all patients.

### 2.2. Surgical Technique

All procedures were performed using a 0.9% saline solution and a continuous flow 26 Ch. resectoscope (Karl Storz, Tuttlingen, Germany), mounted with a 12° optic and with a separate operative channel for the fiber. Laser energy was transmitted using an 800-micron front-firing laser fiber. Enucleation was achieved using a continuous-wave thulium laser (RevoLix DUO 120W, LISA Laser products, Katlenburg, Germany). The laser power setting was 90 W. Morcellation was carried out with the Piranha morcellator using a dedicated double-flow nephroscope (Richard Wolf, Knittlingen, Germany). The surgical technique has already been fully described [17]. To summarize, the approach is a 2-lobe enucleation. First, the median lobe is enucleated, whereas the lateral lobes are dissected and enucleated en bloc afterward. After median lobe enucleation, the dissection begins at the prostatic apex level at 4 o’clock, enucleating the left lobe in an anticlockwise direction toward the bladder neck up to the 12 o’clock position. At this level, enucleation is carried out on the right lobe toward 9 o’clock following the surgical capsule plane. Finally, an incision is made at 8 o’clock, and enucleation is completed proceeding clockwise toward 9 o’clock.

### 2.3. Statistical Analysis

Continuous variables were reported as either their mean and standard deviation or median and interquartile range based on their distribution (assessed using the Shapiro–Wilk test). Patients were divided into four groups according to AP/AC treatment (aspirin, clopidogrel, ticlopidine, and LMWH). Comparison of variables between groups was performed via one-way ANOVA or the Kruskal–Wallis equality-of-populations rank test, according to their distribution. Categorical variables were expressed as absolute numbers and percentages and analyzed by Pearson’s Chi-squared test. Two different outcomes were selected for the multivariate analysis: the probability of developing at least one complication (Clavien of at least 1 or above) and the postoperative length of stay (in days). Logistic or Poisson regression models were built in order to estimate the association between each type of antithrombotic agent and the study outcomes, while taking into account all potential confounders considered in the analysis. Finally, functional outcomes (IPSS, PSA, Qmax, and QoL) were compared at each study point (baseline, and 1, 6, and 12 months after surgery) according to the type of antithrombotic regimen by using the mean and standard deviation or the median and interquartile range based on their distribution. Statistical significance was tested with a one-way ANOVA or a Kruskal–Wallis test as appropriate. The original data from this study are available at Mendeley Data (https://data.mendeley.com/datasets/jnj2c54njg/2).

## 3. Results

Between January 2015 and December 2019, 598 patients underwent the procedure in our Department. Among these, 88 (14.7%) patients met the inclusion criteria and were included in the analysis. Table 1 shows patient demographics and characteristics. The mean age was 74.7 ± 6.1 years, with no statistically significant difference among the four groups. The median PV was 66.5 mL (range 25–180 mL). Patients on aspirin and LMWH were predominant (44.3% and 39.8%, respectively), 69.3% of the patients had an ASA score of 3, and preoperative indwelling catheter was present in 23.9%.

The median operative time was 65 min. No patients were converted to TURP or OP due to intraoperative complications. Blood loss at 24 h was comparable in all groups. Median catheterization length was 2 days and was significantly shorter in patients on aspirin and LMWH (*p* = 0.035). Median postoperative stay was 3 days in all groups. Overall, 23 (26.1%) patients had at least one early complication, and 4 had two (4%). Table 2 illustrates early complications and their management. Most of the complications were mild-to-moderate (74%). Eight bleeding complications occurred in 6/88 (6.8%) patients: 4 patients required prolonged bladder irrigation, 2 required blood transfusions, 1 required cystoscopy for bleeding control, and 1 required a suprapubic cystostomy for blood clot evacuation. Systemic absorption of irrigating fluid occurred in 4/88 (4.5%) patients. Among these men, two were admitted to the intensive care unit and required secondary morcellation. Acute cardiovascular events occurred in 2/88 (2.3%) patients. No patients died within 30 days of discharge.

Logistic and Poisson regressions found no correlation between the antithrombotic regimen and both the occurrence of at least one complication and the length of postoperative stay (Table 3 and Table 4). Nevertheless, logistic regression highlights that the lower the level of bleeding, the less likely patients were to develop complications.

Seventy-eight and sixty-eight men reached 6- and 12-month follow-ups respectively. Late complications occurred in 3/78 (3.8%) patients (1 optical urethrotomy and 1 bladder neck incision for stenosis; 1 acute myocardial infarction). One patient died due to metastatic transitional cell carcinoma of prostatic ducts.

Table 5 provides functional results. All follow-up visits showed a significant improvement in all urinary parameters (IPSS, QoL, and Qmax) compared to baseline, without a statistically significant difference among the groups, except for QoL at 6-month follow-up. A significant median PSA decrease 12 months after surgery demonstrated an excellent enucleation efficiency that was similar in all groups.

## 4. Discussion

BPH is an aging-related process, and LUTS related to BPH constitute a considerable disease burden in men older than 70 years, with a prevalence of 80% in this population [18,19]. Age, smoking, hypertension, type II diabetes, hyperlipidemia, central obesity, and sedentary lifestyle are well-known risk factors associated with both BPH/LUTS and cardiovascular diseases [18,20]. With the aging of the population, urologists are facing more men who require BPH surgery without interrupting AP/AC medications. Thulium:yttrium-aluminum-garnet laser (TL) has recently been introduced in clinical practice for treating BPH [21]. TL operates in a continuous-wave mode at 1940–2013 nm wavelength and has an optical penetration of only 0.2 mm, which permits high energy density, leading to a smooth incision as well as rapid prostatic tissue coagulation and vaporization [22]. These physical characteristics make TL an appealing energy for endoscopic enucleation of the prostate in patients at high risk of bleeding.

Hauser et al. first demonstrated the safety of ThuVEP in 39 men at high risk of bleeding [14]. The mean PV was 50.3 mL, surgical time 92 min, and postoperative stay 4.8 days. Of the patients, 67.4% had an ASA score ≥ 3. Two patients (5.2%) required re-catheterization due to blood clots, and only one patient (2.6%) required a blood transfusion. No data were available regarding major complications and reoperation rate. Netsch et al. reported their experience in 56 patients on oral AC/AP treated in two centers [13]. All patients were at high cardiovascular risk, with a median ASA score of 3. The median PV was 50 mL. Five (8.9%) patients required early reoperation due to hematuria (4) or residual tissue (1), and four (7.1%) needed blood transfusions. There was no data on major acute cardiovascular events. Satisfactory functional results were reported in both series. Our study was consistent with the data showing that men on chronic AC/AP medications had a high comorbidity burden; indeed, 63% of our patients were ASA score ≥ 3. Furthermore, most of the complications in our series were also low-to-moderate (74% Clavien ≤ 2), with only two acute cardiovascular events, 4.5% blood clot retention episodes, and 2.2% transfusions and reoperations for bleeding control. These results were comparable with HoLEP but lower than TURP [23,24]. Nevertheless, three (3.3%) patients in our series required reoperation for secondary morcellation and residual prostatic tissue, and a possible explanation for that might be the impaired vision related to bleeding during enucleation.

Bach et al. evaluated 2648 patients who underwent PVP, ThuVEP, and TURP in real-life settings in four high-volume centers [15]; 455 (17.2%) men were operated on whilst undergoing AC/AP treatments. The authors showed that the median postoperative stay was longer in patients undergoing TURP (4 days) compared to ThuVEP (3 days) and PVP (2 days). Patients on AP/AP had a longer stay in all groups (+0.6 days). Among patients on AC/AP who required reoperation for bleeding, TURP and ThuVEP had the same rate (5% and 5.5% respectively), whereas PVP had a rate of 2.5%. Sixty-four (4.2%) patients required blood transfusions, and the transfusion rate was 4.2% in patients under AC/AP and 2.9% in the remaining, with the lowest rate being in the PVP group (0.6%) and the highest in ThuVEP (3.3%). Interestingly, the author found that an increase in PV and operating time led to an increase in the transfusion rate in TURP and ThuVEP. Our postoperative stay (3 days) and transfusion rate (2.2%) were in line with this large series.

To the best of our knowledge, our study was the first to demonstrate that there was no difference in complication rate and functional outcomes after ThuVEP, regardless of AP regimens. Patients on LMWH also demonstrated comparable results. These findings might suggest that there is no need to switch patients on AP to LMWH in clinical practice.

Aspirin should be maintained in the provision of surgery and the postoperative period in patients at risk for coronary artery disease, because its withdrawal increases the risk of cardiovascular events threefold [4]. Our study supports this finding. During the investigation period, only one patient had an acute myocardial infarction, 8 weeks after surgery. Maintaining AC/AP medications was safe and protected the majority of patients from thromboembolic events, with acceptable postoperative morbidity. Only one patient died, as a result of metastatic transitional cell cancer which was diagnosed in morcellated prostatic tissue. Therefore, no patients died due to surgery-related reasons.

Regarding functional results, the present series confirmed that ThuVEP provided immediate LUTS relief, together with significant QoL and Qmax improvements that persisted for 12 months after surgery. The one-year median PSA decreased by 70.4% compared to the baseline value and confirmed the complete removal of the adenoma. These results are in line with the current large series of ThuVEP and HoLEP [25,26].

Our study has some limitations. First of all, four surgeons with different surgical experience performed the procedures. However, the patients were from a single high-volume center and all procedures were carried out following a standardized approach, while pre- and postoperative patient-management was standardized, depicting a real-life setting. Secondly, no patients were operated on oral AC, however, bridging anticoagulation has been demonstrated to be safe in patients at high risk of thromboembolism following discontinuation of AC [5]. Furthermore, oral AC therapy was resumed in all patients as soon as hematuria subsided. Third, patients on clopidogrel and ticlopidine were few in number, and this could limit the generalization of our results in larger populations of men taking those medications. Finally, a longer follow-up could highlight a higher number of complications and reoperations.

## 5. Conclusions

This study demonstrated that ThuVEP is a feasible option in high-risk patients on AP and LMWH who require surgical treatment of LUTS/BPH, with acceptable postoperative morbidity, good early functional outcome, and a low incidence of medium-term reoperation rate. Further studies with larger populations and long-term follow-up are needed to confirm these results.

## Figures and Tables

**Table 1 jcm-09-00917-t001:** Patient characteristics according to antithrombotic treatments.

	Total	Aspirin	Clopidogrel	Ticlopidine	LMWH	*p*
*n*	88	39	10	4	35	
Age (years)	74.7 ± 6.1	74.0 ± 6.2	74.0 ± 5.8	72.0 ± 8.6	76.0 ± 5.8	0.396
At least 1 complication	23 (26.1%)	8 (20.5%)	5 (50.0%)	1 (25.0%)	9 (25.7%)	0.309
**ASA score**						**0.049**
**2**	**25 (28.4%)**	**12 (30.8%)**	**2 (20.0%)**	**2 (50.0%)**	**9 (25.7%)**	
**3**	**61 (69.3%)**	**27 (69.2%)**	**8 (80.0%)**	**1 (25.0%)**	**25 (71.4%)**	
**4**	**2 (2.3%)**	**0 (0.0%)**	**0 (0.0%)**	**1 (25.0%)**	**1 (2.9%)**	
Indwelling catheter	21 (23.9%)	6 (15.4%)	2 (20.0%)	1 (25.0%)	12 (34.3%)	0.293
**Catheterization length (days)**	**2.0 (1.0)**	**2.0 (1.0)**	**2.5 (2.0)**	**3.0 (1.5)**	**2.0 (1.0)**	**0.035**
Prostate volume (mL)	66.5 (38.5)	66.0 (31.0)	70.5 (50.0)	85.5 (43.0)	65.0 (45.0)	0.756
Surgical time (min)	65.0 (25.0)	70.0 (20.0)	47.5 (25.0)	52.5 (18.5)	60.0 (25.0)	0.115
ΔHb (24 h)	−1.3 (1.0)	−1.0 (0.8)	−1.5 (0.7)	−1.5 (0.5)	−1.3 (1.4)	0.507
**Postoperative stay (days)**	**3.0 (0.0)**	**3.0 (1.0)**	**3.0 (1.0)**	**3.0 (3.5)**	**3.0 (1.0)**	**0.011**

Values are presented as *n* (%) or mean ± SD or median (IQR). LMWH: low-molecular-weight heparin. ASA: American Society of Anesthesiologists. ΔHb: decrease in hemoglobin. Bold value: statistically significant.

**Table 2 jcm-09-00917-t002:** Early complications and their management according to the modified Clavien classification system [16].

Grade	Complication, *n* (%)	Management
**1**	Blood clot retention, 4 (4.5%) Acute urinary retention, 4 (4.5%) Lower UTI, 4 (4.5%) Persistent postoperative headache, 1 (1.1%)	Prolonged bedside bladder irrigation with clot evacuation Bedside re-catheterization Antibiotics Extended bed rest, CT scan of the brain
**2**	UTI with signs of bacteremia, 1 (1.1%) Hematuria with anemia, 2 (2.2%) Dyspnea due to irrigating fluid absorption, 3 (3.4%) Atrial fibrillation, 1 (1.1%)	Antibiotics Blood transfusion Oxygen and diuretics Pharmacologic cardioversion
**3a**	Acute urinary retention due to residual prostatic tissue, 1 (1.1%) Residual prostatic tissue, 2 (2.2%)	TURP in spinal anesthesia Reoperation for secondary morcellation
**3b**	Persistent hematuria, 1 (1.1%) Persistent hematuria, 1 (1.1%)	Cystoscopy for bleeding control under general anesthesia Suprapubic cystostomy for blood clot evacuation in general anesthesia
**4a**	Pulmonary edema to irrigating fluid absorption, 1 (1.1%)	Admission to the intensive care unit
**4b**	Heart failure with pulmonary edema, 1 (1.1%)	Admission to the intensive care unit

TURP: transurethral resection of the prostate. CT: computed tomography. UTI: urinary tract infection.

**Table 3 jcm-09-00917-t003:** Logistic regression (independent variable: at least one complication).

	OR	95% CI
Anticoagulant type (ref. Aspirin)		
Clopidogrel	1.59	0.22–11.38
Ticlopidine	0.26	0.00–14.00
LMWH	0.76	0.19–3.13
Age	1.03	0.91–1.16
ASA Score (ref. 2)		
3	1.27	0.27–5.87
4	1.87	0.03–129.00
Indwelling catheter (ref. no)	0.80	0.13–4.95
**Catheterization length**	**3.36**	**1.36–8.29**
Prostate volume	1.01	0.98–1.04
Surgical time	1.01	0.98–1.06
**ΔHb**	**0.47**	**0.22–0.99**
_constant	0.00	0.00–1.43

Bold value: statistically significant.

**Table 4 jcm-09-00917-t004:** Poisson regression on postoperative stay.

	β	*p*
Anticoagulant type (ref. Aspirin)		
Clopidogrel	0.09	0.672
Ticlopidine	0.25	0.417
LMWH	0.06	0.693
At least 1 complication (ref. no complications)	0.08	0.621
Age	0.01	0.634
ASA Score (ref. 2)		
3	0.11	0.504
4	0.50	0.201
Indwelling catheter (ref. no)	−0.06	0.706
**Catheterization length**	**0.14**	**0.000**
Prostate volume	0.00	0.850
Surgical time	0.00	0.568
ΔHb	−0.04	0.632
_constant	−0.03	0.972

Bold value: statistically significant.

**Table 5 jcm-09-00917-t005:** Functional results at follow-up visits.

	Total	Aspirin	Clopidogrel	Ticlopidine	LMWH	*p*
IPSS_pre ^1^	23.3 ± 4.8	23.6 ± 5.3	23.4 ± 6.7	27.0 ± 2.3	22.5 ± 3.7	0.331
IPSS_1 month ^1^	5.0 (5.0)	5.0 (5.0)	6.0 (7.0)	7.0 (4.5)	5.0 (2.0)	0.781
IPSS_6 months ^2^	3.0 (3.0)	3.0 (3.0)	2.0 (2.0)	2.0 (4.0)	3.5 (3.0)	0.443
IPSS_12 months ^3^	2.0 (2.5)	2.0 (2.0)	2.0 (2.0)	1.5 (2.5)	3.0 (2.0)	0.149
QoL_pre ^1^	4.2 ± 0.9	4.1 ± 0.9	4.4 ± 0.7	4.5 ± 1.3	4.1 ± 0.9	0.660
QoL_1 month ^1^	1.7 ± 1.0	1.6 ± 1.1	1.6 ± 0.8	1.0 ± 0.8	1.9 ± 1.1	0.428
**QoL_6 months ^2^**	**1.3 ± 1.0**	**1.3 ± 0.9**	**0.8 ± 0.7**	**0.5 ± 1.0**	**1.6 ± 1.0**	**0.033**
QoL_12 months ^3^	1.0 (1.0)	1.0 (1.0)	1.0 (0.0)	0.5 (1.0)	1.0 (1.0)	0.344
Qmax_pre ^1^	8.4 ± 2.7	8.9 ± 2.6	7.8 ± 2.2	9.4 ± 3.2	7.8 ± 2.9	0.323
Qmax_1 month ^1^	19.9 (9.8)	20.0 (12.1)	19.4 (7.4)	20.5 (4.3)	19.8 (8.0)	0.938
Qmax_6 months ^2^	20.1 (7.4)	20.1 (10.2)	20.0 (1.9)	20.4 (3.9)	21.0 (7.4)	0.935
Qmax_12 months ^3^	21.6 (6.7)	21.8 (9.3)	20.6 (5.2)	22.5 (3.5)	20.4 (5.2)	0.859
PSA_pre ^1^	2.7 (3.3)	2.9 (2.6)	2.3 (4.8)	3.4 (6.0)	2.5 (2.4)	0.748
PSA_12 months ^3^	0.8 (0.6)	0.8 (0.6)	0.7 (0.2)	1.0 (1.0)	0.7 (0.8)	0.480

Values are presented as mean ± SD or median (IQR). PSA: prostate-specific antigen. IPSS: International Prostate Symptom Score. QoL: quality of life. Qmax: maximal urine flow rate. ^1^
*n* = 88; ^2^
*n* = 78; ^3^
*n* = 68. Bold value: statistically significant

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
