# Peer review of "Thulium Laser Vapoenucleation of the Prostate (ThuVEP) in Men at High Cardiovascular Risk and on Antithrombotic Therapy: A Single-Center Experience"

_jcm, 2020, doi:10.3390/jcm9040917_

Round 1

Reviewer 1 Report

Thank you for the opportunity to review the presented work of Castellani et al. 

"Thulium laser vapoenucleation of the prostate (ThuVEP) in men at high cardiovascular risk and on antithrombotic therapy: a single center experience."

General.

Interesting topic, very much debated in the past, on of the mayor concerns in daily practise

Study type retrospective case series (It would be prospective - if this question would be the defining criteria for setting up the database) - must be mentioned somewhere to depict quality of data. Comment / in gerneal there is not better data than that.

Relevant "topics" are clopidrogel (10 patients) and Ticlopidin (4 patients) among 88 patients

The problem is that a lot of data exists for ASS and LMH, data is lacking for Cloipidrogel, Warfarin and NOAKs (30 day morbidity).

Minor  

Page 2

46 Both the EAU and American Urological Association guidelines recommend that endoscopic
47 enucleation of the prostate with holmium (HoLEP) or thulium laser support can also be offered in 48 patients who are at higher risk of bleeding [10,11]. 

Some words are ,missing "Thulium vapoenucleation and transuirethral anatomical enucleation with ..."

Page 2

66 Novel oral anticoagulant agents and warfarin were stopped 2–3
67 days before surgery and bridged with low-molecular-weight heparin (LMWH) [5]. 

on the base of CHADVASC?

 in General

Author Response

Reviewer #1

- Thank you for the opportunity to review the presented work of Castellani et al. 

"Thulium laser vapoenucleation of the prostate (ThuVEP) in men at high cardiovascular risk and on antithrombotic therapy: a single center experience."

General.

Interesting topic, very much debated in the past, on of the mayor concerns in daily practise

RESPONSE

We want to thank the referee for this nice comment on our paper.

- Study type retrospective case series (It would be prospective - if this question would be the defining criteria for setting up the database) - must be mentioned somewhere to depict quality of data. Comment / in general there is not better data than that.

RESPONSE

We want to thank the referee for this nice comment. We agree with her/him that the present study is retrospective, even if the data were prospectively collected. This has been mentioned in Material and Methods as follows (line numbers 60-61): "We retrospectively analyzed data from all men who consecutively underwent ThuVEP at our center."

- Relevant "topics" are clopidrogel (10 patients) and Ticlopidin (4 patients) among 88 patients

The problem is that a lot of data exists for ASS and LMH, data is lacking for Cloipidrogel, Warfarin and NOAKs (30 day morbidity).

RESPONSE

We want to thank the referee for this helpful comment. We agree with her/him that patients on clopidogrel and ticlopidine were few. This limitation has been added in Discussion as follows (line numbers 225-226): "Third, patients on clopidogrel and ticlopidine were few, and this could limit the generalization of our results in larger populations."

- Minor  

Page 2

46 Both the EAU and American Urological Association guidelines recommend that endoscopic
47 enucleation of the prostate with holmium (HoLEP) or thulium laser support can also be offered in 48 patients who are at higher risk of bleeding [10,11]. 

Some words are ,missing "Thulium vapoenucleation and transuirethral anatomical enucleation with ..."

RESPONSE

We want to thank the referee for this helpful comment. The term transurethral anatomical was used instead of endoscopic (line 47).

- Page 2

66 Novel oral anticoagulant agents and warfarin were stopped 2–3
67 days before surgery and bridged with low-molecular-weight heparin (LMWH) [5]. 

on the base of CHADVASC?

RESPONSE

We want to thank the referee for this comment. Warfarin and novel oral anticoagulant agents were stopped before surgery and bridged with LMWH in all patients despite their CHA2DS2-VASc score. The decision to stop anticoagulants was made after cardiological/neurological consultation in all patients.

Reviewer 2 Report

The authors evaluated the safety and effectiveness of Thulium laser vapoenucleation of the prostate (ThuVEP) for benign prostatic hyperplasia in patients on chronic antithrombotic medications. They concluded ThuVEP was a feasible surgical option in high-risk patients on antithrombotic regimens.

This is an interesting and impressed report. The contents are concise and easy to read.

A comparative study was conducted between antithrombotic medications.

Authors had better to show the results of control group who did not take such medication, if they want to emphasize that there was no need to discontinue the drug.

Author Response

Reviewer #2

The authors evaluated the safety and effectiveness of Thulium laser vapoenucleation of the prostate (ThuVEP) for benign prostatic hyperplasia in patients on chronic antithrombotic medications. They concluded ThuVEP was a feasible surgical option in high-risk patients on antithrombotic regimens.

This is an interesting and impressed report. The contents are concise and easy to read.

A comparative study was conducted between antithrombotic medications.

RESPONSE

We want to thank the referee for this nice comment on our paper.

- Authors had better to show the results of control group who did not take such medication, if they want to emphasize that there was no need to discontinue the drug.

RESPONSE

We want to thank the referee for this valuable comment on our paper. We agree with her/him that a control group would be necessary to emphasize that there was no need to discontinue the drug. Nevertheless, the aims of the present study were to assesses the complication of ThuVEP in men on chronic antiplatelet drugs and to investigate the relationship between different antithrombotic regimens. The results of our study showed some important practical implications in managing such patients in daily practice (no 30-day mortality, low-grade early morbidity, and no difference among antithrombotic drugs). Furthermore, we have already demonstrated in a previous publication in an elderly population that complications and outcomes were similar in men not taking antithrombotic drugs compared to those under such medications (Castellani D et. al. Are Outcomes of Thulium Laser Enucleation of the Prostate Different in Men Aged 75 and Over? A Propensity Score Analysis. Urology. 2019 Oct;132:170-176. doi: 10.1016/j.urology.2019.06.025.). Moreover, the results of the present study are in line with those of the reference paper on ThuVEP (Gross et al. Complications and early postoperative outcome in 1080 patients after thuliumvapoenucleation of the prostate: results at a single institution. Eur Urol. 2013 May;63(5):859-67. doi: 10.1016/j.eururo.2012.11.048.).

Round 2

Reviewer 2 Report

Authors The authors seem to have responded appropriately.